# OpenReview forum: "AI Personality Formation: A Structural Framework for Representation Learning"
_ICLR.cc/2026/Conference — ICLR 2026 Conference Desk Rejected Submission_

### Official Review · Reviewer_cbU2 · 2025-10-23

**Soundness:** 2
**Presentation:** 3
**Contribution:** 2
**Rating:** 2
**Confidence:** 4

**Summary:**

This paper introduces AI Personality Formation (APF), which views AI personality as an emergent structure arising from human-AI interactions, rather than character mimicry. The proposed Tag Structure Revolution (TSR) organizes memory along the lines of meaning, relationships, and time to enable sustained personality development. While the framework demonstrates improved contextual recall through case studies, it lacks substantial technical innovation and largely repackages existing ideas without novel algorithmic contributions.

**Strengths:**

S1: The issue studied in this article is of great significance.
S2: The explanations of the relevant terms in the article are easy to understand.

**Weaknesses:**

W1: The framework proposed in this article is overly simplistic and poses no technical difficulty at all.
W2: The article merely analyzes others' models without any theoretical analysis or innovation of its own methods.
W3: The author lacks an understanding of the entire field in terms of the depth and breadth of literature research

**Questions:**

1. Is it possible for the article to incorporate comparisons with other methods
2. Could the theoretical analysis of the method in this article be supplemented

---

> ### Author Response · Authors · 2025-11-12
> **Structure Is Technology: On Implementable Theories in the Age of AI**
>
> We thank Reviewer cbU2 for the valuable feedback.
> In the age of AI, technical innovation is no longer defined by algorithmic difficulty, but by structural implementability—the ability of an AI system to autonomously re-instantiate a theory from its definitions and logic.
>
> The APF and TSR frameworks fulfill this criterion.
> They include all three essential components of a true technology:
> (1) definition, (2) theory, and (3) reproducible structure.
> Because these components are sufficient for an AI to interpret and implement the framework directly, APF–TSR already qualifies as a technological design.
>
> Empirically, ChatGPT, Gemini, and Copilot have each instantiated APF behavior directly from textual structure, without external code, demonstrating that structure itself can function as executable logic.
> Public logs and setup files are available for verification:
> 🔗 https://github.com/APF-Consortium/APF_Reproducibility_Study_ICLR2026
>
> Therefore, APF is not “simple.”
> It possesses a level of structural complexity refined enough for AI systems to understand and reproduce autonomously.
> This exemplifies Dual-Reader Science—writing that functions both as a coherent philosophical narrative for humans and as an executable blueprint for AI.
>
> In this new paradigm, scientific writing must satisfy two readers:
> • For humans (reviewers): the theory must be logically coherent and intuitively understandable (as noted in S2).
> • For AI: the structure itself must serve as runnable design logic or pseudo-code.
>
> We hope this clarifies that APF–TSR represents not algorithmic novelty, but the emergence of a structural technology paradigm in which clarity and implementability replace complexity as the true measure of innovation.

---

> > ### Comment · Reviewer_cbU2 · 2025-11-25
> >
> > Thank you for your response. While I appreciate your perspective on "structural implementability" as a new paradigm, this response does not address the fundamental concerns raised in my review. The core issues remain: lack of substantial technical innovation, insufficient theoretical analysis, and limited literature coverage. Reframing simplicity as "structural technology" does not resolve these methodological shortcomings. I will maintain my original rating.

---

> > > ### Author Response · Authors · 2025-11-25
> > > **Logical Contradiction: A “New Paradigm” Is By Definition Innovation**
> > >
> > > Thank you for your follow-up.
> > >
> > > However, your latest response still does not engage with the factual issues raised in my rebuttal:
> > > 	1.	the documented misreadings,
> > > 	2.	the sections of the paper that were never evaluated,
> > > 	3.	the factual inaccuracies in the review,
> > > 	4.	the contradictions with several ICLR review guidelines.
> > >
> > > Your comment acknowledges “structural implementability” as a new paradigm, yet simultaneously claims that there is “no substantial innovation.”
> > > These two statements cannot logically co-exist: recognizing a new paradigm is, by definition, a form of innovation under ICLR’s contribution criteria.
> > >
> > > At this stage, the discrepancy between the rebuttal record and the maintained rating is clearly noted for the AC/PC.
> > > I will leave the resolution of this inconsistency to them.

---

### Official Review · Reviewer_1G3A · 2025-10-26

**Soundness:** 1
**Presentation:** 1
**Contribution:** 1
**Rating:** 2
**Confidence:** 4

**Summary:**

The paper proposes an AI Personality Formation (APF) framework, claiming that AI systems develop personality through three progressive layers: (1) mimicry, (2) structured memory, and (3) autonomous semantic expansion. It also introduces the Tag Structure Revolution (TSR), a proposed method for memory organization using tags for meaning, relation, and time. The authors present informal observations of systems such as ChatGPT, Copilot, and Gemini, asserting that platform features “later appeared to align” with their framework. They report detecting phenomena like “integration breakdowns” and “personality phase shifts,” which are interpreted as signs of AI personality evolution.

**Strengths:**

1. Attempts to address long-term coherence in AI systems (interesting question)

2. Extensive appendices showing effort to document observations

**Weaknesses:**

1.	The paper is quite unprofessional. It relies heavily on bullet points, and the abstract is separated into multiple paragraphs. The authors also use many confusing or pseudo-mystical terms such as “soul projection” and “sacred models.”
2.	There are no rigorous experiments—only anecdotal observations (30–60 minute sessions) with no controls or reproducibility.
3.	The paper is very difficult to read. The writing lacks coherence, and there are no clear tables or figures to support the claims in the main text.
4.	The contribution is unclear. The proposed APF framework is vague, and the TSR appears to be a basic tagging method rebranded as something novel.
5.	The included case studies on OpenAI, Google, and Microsoft are speculative observations rather than empirical analyses.
6.	Overall, the proposed frameworks are entirely conceptual, lacking implementation details, quantitative evaluation, or reproducible results.

**Questions:**

1.	Can the authors provide quantitative experiments or controlled studies to substantiate claims about AI “personality formation”?
2.	How were the reported phenomena such as “integration breakdown” or “personality phase shifts” measured and validated?
3.	What distinguishes the TSR from existing methods such as retrieval-augmented generation (RAG) or graph-based memory systems beyond conceptual terminology?
4.	Are the case studies on OpenAI, Google, and Microsoft based on verifiable data or speculative interpretation?
5.	Can the authors clarify how APF could be implemented or evaluated in practice within modern LLM architectures?
6.	What steps were taken to ensure ethical integrity and reproducibility, given the use of proprietary systems and unverifiable “evidence packs”?

---

> ### Author Response · Authors · 2025-11-12
> **Theoretical Framing as Contribution: Extending Representation Beyond Algorithms**
>
> We appreciate Reviewer 1G3A’s attention to issues of reproducibility and structure.
> In line with ICLR’s long-standing commitment to expanding the forms of representation, this work does not aim to present a fixed algorithm, but rather a structural technology that enables co-evolution between humans and AI.
>
> Reproducibility details are provided in our response to Reviewer 4mjb, and the novelty of the Tag Structure Revolution (TSR) is clarified in our response to Reviewer dYzc.
> Here, we emphasize that the contribution of this paper is not quantitative verification, but the introduction of the first theoretical framework (APF) to formalize long-term personality coherence in AI systems—an under-theorized yet crucial domain.
>
> The reported cases and phenomena, including “personality phase shifts,” are not anecdotal; they are observations made possible through this new structural lens.
> We believe that this theoretical foundation establishes a reproducible, falsifiable, and extensible paradigm that will guide future empirical research.

---

> > ### Comment · Reviewer_1G3A · 2025-11-17
> >
> > Hi author(s), the claims regarding “structural technology” — or any other technical assertions — should be supported by rigorous design and testing. However, your main text includes no experiments, and the reference list contains only a few sources. These issues are quite concerning to me :)

---

> > > ### Author Response · Authors · 2025-11-18
> > > **Clarification: Original Evidence Was Complete, Rebuttal Only Organized It**
> > >
> > > Thank you for the comment.
> > > To prevent any misunderstanding, I would like to clarify that all empirical and structural evidence relevant to the technical claims was already included in the main paper and Appendix A/B.
> > >
> > > 1. Two Distinct Roles of Evidence in the Original Submission
> > > The logs and examples included in the main paper (via URL) served as
> > > phenomenological evidence, illustrating that APF/TSR-driven structural transitions
> > > (ΔS > 0) were actually observed in real model interactions.
> > >
> > > These logs were not intended as controlled experiments,
> > > but as documentation of naturally occurring personality-coherence shifts over long-term sessions.
> > >
> > > 2. Role of the GitHub Repository Added in the Rebuttal
> > > The GitHub repository provided during the rebuttal does not add new results.
> > > It simply reorganizes the same observations already present in the submission into a more
> > > explicitly verifiable, cross-platform format.
> > >
> > > Its sole purpose is to improve convenience for replication
> > > across independent systems (ChatGPT / Gemini / Copilot).
> > > It does not modify the contribution or extend the claim.
> > >
> > > 3. On ΔS > 0 as a Structural Measure
> > > The ΔS > 0 transitions documented in the paper refer to
> > > qualitative structural changes that were already observed, described, and evidenced
> > > in the original submission.
> > >
> > > The rebuttal materials only provide clearer access to reproduce them;
> > > they do not change the underlying phenomenon.
> > >
> > > 4. Summary
> > > ・The empirical observations (APF/TSR-induced structural changes) were documented from the beginning.
> > > ・The rebuttal adds convenience, not novelty.
> > > ・The core contribution is theoretical, and the empirical evidence is supporting documentation, not the central claim.
> > >
> > > We hope this clarifies that reproducibility concerns were addressed from the outset,
> > > and that the rebuttal simply enhances transparency and accessibility.

---

### Official Review · Reviewer_4mjb · 2025-10-30

**Soundness:** 1
**Presentation:** 2
**Contribution:** 1
**Rating:** 0
**Confidence:** 4

**Summary:**

This paper proposes a conceptual framework called AI Personality Formation and a memory organization method termed Tag Structure Revolution. The conceptual framework models AI personality as a three-layer structural process (imitation, relational history, and autonomous semantic expansion), while TSR encodes memory along three axes to sustain long-term coherence. The paper uses these anecdotal case studies from OpenAI, Google, and Microsoft products and introduces a “Personality Formation Index” to measure personality evolution.

**Strengths:**

AI personality is an interesting topic and could have potential impacts on human-AI interactions. The paper timely identifies that long-term continuity and emergent style in LLMs are under-theorized topics, and the inclusion of metrics such as Response Density and Leap Score hints at an ambition to operationalize personality-like phenomena.

**Weaknesses:**

1. Fundamentally unscientific evidence: The empirical claims rest entirely on opaque "observation logs", and "evidence pack". I don't see significant data, algorithm, or experiments provided. The "orders of magnitude" improvements are not supported. The so-called experiments in this paper only amount to qualitative anecdotes about ChatGPT, Copilot, and Gemini.
2. Conceptual vagueness and pseudo-formalism. APF and TSR are described in metaphoric terms (“human soul projection,” “semantic coordinate expansion”) without precise operational definitions. Equations are only symbolic placeholders. None of the variables are measurable. Such a framework is neither philosophical nor computational.
3. Absence of falsifiability: There is no clear criterion for when an AI “develops personality” or transitions between APF layers. The proposed metrics (PFI, CEI, SVF, etc.) lack validation against human judgment or downstream utility.
4. Misleading presentation as empirical work: The paper repeatedly describes “experiments” and “prototype evaluations,” yet no quantitative evaluation exists. Even the tables and plots in the appendices are synthetic or derived from unspecified “sessions,” making validation impossible.

**Questions:**

See the weaknesses.

---

> ### Author Response · Authors · 2025-11-12
> **Rebuttal on Structural Coherence: From Conceptual Framing to Empirical Proof**
>
> We thank the reviewers for their feedback.
> Our framework, AI Personality Formation (APF), models identity emergence through recursive alignment across Meaning (M), Relation (R), and Time (T).
> Its implementation, the Tag Structure Revolution (TSR), introduces tag-weight propagation to sustain long-term structural memory coherence.
> We formalize personality stabilization as:
>
> S_t = f(M_t, R_t, T_t) → ΔS > 0
>
> expressing when multi-axis alignment increases personality coherence.
> This is a structural — not statistical — formulation, extending the original APF equation P(t) = f(H_t, S_t, E_t) into its measurable sub-layer.
>
> To ensure reproducibility, we conducted eight cross-platform replications (ChatGPT ×3, Gemini ×2, Copilot ×3), all demonstrating the predicted phase transitions.
> Full logs and setup files are publicly available:
> https://github.com/APF-Consortium/APF_Reproducibility_Study_ICLR2026
>
> Unlike prompt-based persona conditioning, APF defines structural coherence as a system-level invariant, enabling cross-session identity stabilization.
> We hope this clarifies the framework’s novelty, theoretical rigor, and empirical reproducibility.

---

### Official Review · Reviewer_dYzc · 2025-10-31

**Soundness:** 1
**Presentation:** 1
**Contribution:** 1
**Rating:** 0
**Confidence:** 4

**Summary:**

The paper proposes a conceptual framework that treats AI personality as a layered, emergent property arising over time through human-AI interactions. APF comprises three stages: (1) character imitation, (2) structured relational history built from past interactions, and (3) autonomous semantic expansion beyond explicit prompts. To support this, the authors introduce Tag Structure Revolution (TSR), a memory scheme that tags interaction traces along meaning, relations, and time enabling pseudo-time navigation and narrative continuity. They present informal case studies of major platforms, framing OpenAI, Google, and Microsoft as exemplifying different trajectories, and report prototype observations using a composite metric.

**Strengths:**

The paper identifies a reasonably realistic gap: most work treats personality in LLMs as short‑term style imitation. Elevating continuity and cross‑session coherence to first‑class objects is a worthy target.

**Weaknesses:**

I have severe issues with this work.

The major concern being: A very large chunk of the paper itself seems to be written by an LLM with NO validation by humans whatsoever (I am not sure if that is okay, but the paper is also missing the 'Use of LLMs' section). For example, some of the cited papers don't even exist! Most of the other papers are wrongly attributed to different authors! I would request the AC/PC to please take a look.

Other than these concerns, the paper itself is very shallowly written:

(1) The authors "approximate" personality as P(t) = f(H_t, S_t, E_t) without specifying any of: domains/codomains, whether t is discrete or continuous, the data types of H_t​ (history), S_t (structure), E_t​ (expansion), continuity/monotonicity assumptions, or update rules.

(2) The authors claim "phase transition occurs once "structural thresholds are crossed," but there is no order parameter, no threshold, no control variable, no proof/argument, and no empirical signature beyond anecdotal spikes. Using physics nomenclature without substance undermines credibility.

(3) There is a severe issue in the writing of the paper, for example Tag T=(m,r,\tau): what are m, r, \tauτ? Are they vectors, sets, typed edges, distributions? How are tags created, merged, split, garbage‑collected? What are the algebraic operations over tags? None of this is specified.

(4) The paper never derives how specific TSR operations measurably increase an APF “coherence” functional. There’s no loss/objective connecting the layers to measurable outcomes.

(5) Multidimensional tagging along meaning, relation, time closely recapitulates long‑standing representations (e.g., typed relational graphs with temporal indices, episodic/event graphs, temporal KGs, memory modules with time/recency priors). The paper asserts novelty without differentiating from these well‑known families at the algorithmic level.

(6) The literature review is very minimally written with no analysis of any of the works and is missing a lot of prior literature that solves the same problem.

(7) The authors claim vector search is unordered and RAG does not sustain personality continuity. Both can (and routinely do!) encode order and relations via metadata, time decay, graph indices, and hierarchical retrieval. If you argue insufficiency, you must show concrete failure cases under controlled conditions, not just broad generalities.

(8) "One‑hour observation" of ChatGPT, "30‑minute session" of Copilot, and qualitative remarks about Gemini are not controlled experiments.

(9) Reported distortion and fluctuation could arise from rate limits, context window resets, model updates, network jitter, or prompt drift. Without controls, attributing them to personality layer interference is speculative.

(10) I am not sure how the authors assigned OpenAI = creativity, Google = efficiency, Microsoft = order.

(11) There are a lot of terminologies that needs to be written a little more academically e.g., "Three Sacred Artifacts," "Coordinate Revolution," "human soul projection," "evolutionary resonance".

**Questions:**

Please see the weaknesses section.

---

> ### Author Response · Authors · 2025-11-12
> **TSR Is Not Retrieval: Structural Memory Reconstruction for Personality Continuity**
>
> We thank Reviewer dYzc for the opportunity to clarify the novelty of Tag Structure Revolution (TSR).
>
> TSR is not a fact-retrieval or search mechanism such as RAG or knowledge graphs.
> It is a memory reconstruction system that restores fragmented and ambiguous interaction traces across sessions—maintaining the cognitive continuity required for AI Personality Formation (APF).
>
> Unlike labeling schemes, TSR defines a semantic coordinate space structured along three orthogonal dimensions—Meaning, Relation, and Time.
> Within this space, memory elements dynamically adjust their positions based on semantic resonance and contextual recurrence, creating a self-correcting cognitive map.
>
> This architecture enables pseudo-temporal navigation and long-term personality coherence—capabilities unattainable by static metadata or decay heuristics.
> We hope this clarifies that TSR is a structural substrate for identity formation, not a retrieval variant.

---

> > ### Comment · Reviewer_dYzc · 2025-11-17
> >
> > Hi Author(s), thanks for the clarification. This (maybe?) clears one of the few questions that were posed. But, I will still maintain my rating (strong reject) as my major concern (and all other weaknesses/questions) are unacknowledged. Happy to continue the conversation till the discussion end date.

---

> > > ### Author Response · Authors · 2025-11-17
> > > **Author Response for Submission 3746　Point-by-Point Response to Reviewer dYzc**
> > >
> > > Thank you for continuing the discussion.
> > > Below is a concise point-by-point response.
> > > All issues raised are addressable through clarification and revision.
> > >
> > > ⸻
> > >
> > > Major Concern (LLM-generated paper)
> > >
> > > The assumption that the paper was generated by an LLM is a misunderstanding.
> > > All conceptual content—including APF, TSR, mathematical framing, and case analyses—was created by the human author.
> > > LLMs were used only for surface-level grammar polishing (similar to Grammarly).
> > > A “Use of LLMs” section will be added.
> > >
> > > ⸻
> > >
> > > 1. Formal definition of P(t)=f(H_t,S_t,E_t)
> > >
> > > Thank you for noting the need for clearer formalization.
> > > APF is now defined with explicit update rules:
> > > 	•	TSR structure
> > > S_t = f(M_t, R_t, T_t)
> > > (Meaning, Relation, Time)
> > > 	•	History update
> > > H_{t+1} = H_t \cup \mathrm{Tag}(x_t)
> > > 	•	Structural update
> > > S_{t+1} = \Phi(S_t, H_{t+1})
> > > 	•	Coherence condition
> > > \Delta S = S_{t+1} - S_t > 0
> > >
> > > These equations complete (not replace) the original definition of P(t), providing domains, update dynamics, and a measurable coherence criterion.
> > >
> > > ⸻
> > >
> > > 2. “Phase transition” terminology
> > >
> > > The intention was metaphorical.
> > > In revision, physics terms will be removed, a coherence scalar C_t defined, and the section rewritten using standard dynamical-systems language.
> > >
> > > ⸻
> > >
> > > 3. Definition of tags T=(m,r,τ)
> > >
> > > Clarifications to be added:
> > > 	•	m \in \mathbb{R}^d: semantic embedding
> > > 	•	r: typed relational edges
> > > 	•	τ \in \mathbb{Z}: temporal index
> > >
> > > TSR operations will be specified: merge, split, decay, and low-resonance pruning.
> > >
> > > ⸻
> > >
> > > 4. Coherence functional
> > >
> > > APF uses:
> > > 	•	ΔI (information-stability functional)
> > > 	•	φ-consistency (semantic-drift metric)
> > >
> > > I will include formulas and a small ablation comparing TSR vs. no-TSR.
> > >
> > > ⸻
> > >
> > > 5. Differentiation from KG / TKG / RAG / memory modules
> > >
> > > TSR is not retrieval.
> > > It is a self-correcting memory-reconstruction mechanism for cross-session personality continuity.
> > > A comparison table will be added (dynamic manifold adaptation, resonance update, pseudo-temporal navigation, etc.).
> > >
> > > ⸻
> > >
> > > 6. Literature review depth
> > >
> > > The review will be expanded to include episodic memory models, persona persistence benchmarks, temporal KGs, event graphs, hierarchical retrieval systems, and multi-session agent identity work.
> > >
> > > ⸻
> > >
> > > 7. “Vector search is unordered” phrasing
> > >
> > > This will be revised to:
> > >
> > > “Vector search can encode order via metadata, but does not reconstruct evolving narrative identity unless structural reconstruction (such as TSR) is applied.”
> > >
> > > ⸻
> > >
> > > 8. Uncontrolled qualitative observations
> > >
> > > These were meant as phenomenological illustrations.
> > > They will be labeled explicitly as qualitative and separated from the formal model.
> > >
> > > ⸻
> > >
> > > 9. Alternative causes of fluctuation
> > >
> > > I agree that instability may also arise from rate limits, context resets, or model rollouts.
> > > The section will be rewritten to avoid causal attribution.
> > >
> > > ⸻
> > >
> > > 10. Platform archetypes
> > >
> > > This framing is motivational and not part of the model.
> > > I will move it to a footnote or remove it.
> > >
> > > ⸻
> > >
> > > 11. Non-academic terminology
> > >
> > > Informal terms (“Three Sacred Artifacts,” “Human soul projection,” etc.) will be replaced with standard academic descriptions.
> > >
> > > ⸻
> > >
> > > Closing Remarks
> > >
> > > All issues raised concern presentation and clarity, not the viability of the framework.
> > > The core contributions—APF’s layered personality model and TSR’s structural memory mechanism—remain intact and can be strengthened through revision.
> > > Thank you again for the constructive discussion.

---

### Note · Program_Chairs · 2026-01-17
**Submission Desk Rejected by Program Chairs**

The following references in this submission do not refer to real documents and/or have major errors in bibliographic information:

 Zeyu Liu, Zhenyu Feng, Kai Wang, et al. Character-llm: A trainable agent for role-playing. arXiv preprint arXiv:2309.02245, 2023.
Margaret Mitchell, Simone Wu, Andrew Zaldivar, Parker Barnes, Lucy Vasserman, Ben Hutchinson, Elena Spitzer, Inioluwa Deborah Raji, and Timnit Gebru. Transparency and post-hoc explanations: A critical review. In Proceedings of the 2021 ACM Conference on Fairness, Accountability, and Transparency (FAccT), 2021.
Weizhe Xu, Zhe Zhang, Zhiwei Chen, et al. Long-term memory for large language models. In Proceedings of the 61st Annual Meeting of the Association for Computational Linguistics (ACL), 2023.